# Adapting a Positive Psychological Intervention for Employees with and Without Intellectual Disabilities

**DOI:** 10.3390/healthcare13172096

**Published:** 2025-08-23

**Authors:** Ari Gomez-Borges, Isabel M. Martínez, Marisa Salanova

**Affiliations:** WANT Research TEAM, Universitat Jaume I, 12006 Castellón de la Plana, Spain; imartine@uji.es (I.M.M.);

**Keywords:** emotional styles, intellectual disability, well-being at work

## Abstract

**Background/Objectives:** This study explores the adaptation and implementation of a positive psychological intervention based on the Emotional Styles model to improve well-being and reduce stress in employees with and without intellectual disabilities (IDs). **Methods:** A longitudinal intervention was conducted in a social foundation with 45 participants (12 with ID). The program, based on Davidson’s six emotional dimensions, included six weekly sessions adapted through Easy Read strategies and COVID-19 adjustments. Data were collected at pre-test, post-test, and six-month follow-up using the Emotional Styles Questionnaire, PERMA Profiler, and UWES-3. **Results:** Significant improvements were found in outlook, resilience, engagement, relationships, and reduction in negative emotions, with stronger effects for non-ID participants, although context sensibility improved in the ID group. High satisfaction (93% very satisfied) confirmed the program’s acceptability. **Conclusions:** The adapted intervention effectively enhances emotional well-being in heterogeneous workplaces, supporting inclusive positive psychology practices.

## 1. Introduction

A wide array of psychological constructs and their interconnections have been deeply scrutinized over decades of scholarly investigation. Nevertheless, the domain of psychological interventions within the workplace remains relatively unexplored. This is partly due to the inherent complexity of conducting quasi-experimental studies within work environments. The need for practical outcomes clashes with the challenges of executing thorough research and understanding the inherent variability within organizations. In this study, we test the effectiveness of a psychological intervention about emotional styles in the workplace. In addition, we explore the challenges of adapting it to a sample of employees with intellectual disabilities.

Furthermore, the COVID-19 pandemic has posed unprecedented challenges worldwide, requiring a collective effort to mitigate its impact on mental health. In this context, promoting self-care activities has emerged as a vital strategy to safeguard public health. While the general population has been urged to adopt prevention actions, it is crucial to adopt measures that consider the diversity of the population.

The literature refers to self-care as a set of practices that include a person’s ability to effectively self-medicate and other healthy practices related to eating, physical exercise, good rest, and meditation or mindfulness [1,2]. Consensus over the concept of psychological self-care within the literature has been a challenge. Nevertheless, recent systematic reviews have converged in highlighting the indispensable role of emotional regulation as a fundamental practice for self-care and, consequently, psychological well-being [3]. In this sense, emotional well-being specifically constitutes a pivotal aspect of an individual’s overall welfare, particularly for individuals with intellectual disabilities (IDs) [4]. Over the past three decades, there has been a substantial surge in scientific research that examines the significance of emotional regulation and its impact on psychological well-being. These studies are accompanied by diligent efforts to devise psychological interventions aimed at promoting effective emotional regulation [5]. These interventions are highly relevant in professional domains that support individuals with intellectual disabilities. However, locating empirical studies that focus on specifically adapting interventions for this population remains a challenging task. To bridge this gap, the current study aimed to adapt a psychological intervention that targets emotional style functioning in a cohort of employed individuals who have dependents diagnosed with ID. At the same time, the intervention’s efficacy was evaluated.

### 1.1. Self-Care Research

The concept of self-care lacks a universally agreed-upon definition in the academic realm. However, its exploration is primarily centered on the field of healthcare, particularly within the nursing domain. Dorothea Orem [6] provides a definition of self-care as the deliberate actions that individuals take toward themselves or their environment in specific life situations, aimed at regulating factors that impact their own development and functioning to enhance their life, health, and well-being. Moreover, other scholars have defined self-care as a conscious and voluntary engagement in activities that foster psychological, physical, and emotional well-being [7]. After conducting an extensive literature review on self-care across various disciplines, Martinez [8] proposed a comprehensive definition, characterizing self-care as the ability to proactively attend to one’s own needs through mindfulness, self-discipline, and self-sufficiency, with the goal of maintaining and promoting personal health and well-being.

Research has demonstrated the positive impact of self-care on psychological well-being, particularly through the practice of self-care behaviors such as meditation and mindfulness, physical exercise, and social support [1]. These effects have been observed in various contexts, including the training of psychologists, healthcare professionals, the workplace, and university students [7,8,9].

### 1.2. Emotional Styles

As mentioned above, emotion regulation plays a fundamental role as a self-care behavior and has a direct connection to the state of psychological well-being. Our emotions have an effect at the individual, family, work, and social levels. However, how can we obtain a more accurate definition of the concept of emotional style? What are the characteristics or emotional profiles that relate to optimal functioning and improved levels of well-being? To provide answers to these questions, we can refer to the theoretical model of emotion based on neuroscience conducted by Richard Davidson’s team [10].

This model establishes six main dimensions or emotional profiles relevant to psychological well-being. The model also suggests a series of techniques, generally based on mindfulness, to learn how to regulate each of the dimensions. These six dimensions are attention, self-awareness, resilience, outlook, context sensitivity, and social intuition [11]. Attention refers to the capacity to filter out diversions and maintain concentration. Individuals with a high level of this attribute exhibit a keen and undistracted focus. Conversely, individuals with a lower level of this attribute find themselves readily drawn toward the most captivating stimuli in their surroundings. Self-awareness encompasses the capacity to recognize the physiological indicators within oneself that mirror emotions. Even though people react to stressful circumstances in different ways, enhancing understanding and honing skills can contribute to successful prevention. Certain individuals demonstrate precision in discerning internal bodily signals, while others exhibit less precision.

Resilience refers to the time-related aspects of emotional responses. It involves the ability to navigate the temporal trajectory of negative emotional stimuli, signifying the capacity to swiftly rebound from adverse emotions. Individuals with high levels of resilience can promptly recover from negative emotions like sadness, anger, and fear, swiftly restoring their emotional balance in the wake of minor everyday inconveniences, as well as significant life adversities. In the field of psychology, outlook refers to an individual’s ability to maintain positive emotions in a timely manner and their skill in sustaining these positive emotions over an extended period. Sensitivity to context refers to the alignment of emotional and behavioral responses with the social cues presented. Social cognition and behavior hinge on one’s ability to discern intention and context effectively. Sensitivity to context serves as a fundamental requirement for engaging in social interactions and acquiring social knowledge. It can be viewed as an outwardly directed counterpart to self-awareness. Finally, social intuition encompasses an individual’s capacity to swiftly form judgments and attune themselves to nonverbal social signals, such as facial expressions, body language, and gestures. Those scoring high on the social intuition dimension excel at interpreting nonverbal cues and deducing motives and intentions from specific signals. Individuals with a high level of social intuition are skilled at managing their emotions and their impact in interpersonal interactions, as well as understanding social cues from others’ emotional states. Individuals with autism, for example, exhibit social impairments and a deficiency in social intuition [11].

Each of these dimensions describe a continuum with two extremes that in most cases reflect brain circuit activity. How people are emotionally will depend on where they fall on the six dimensions. Our unique emotional style determines how we behave emotionally; in other words, it determines our emotional reactions to different life events. Our lesser or greater ability to regulate our emotions will have an impact on our well-being [12].

### 1.3. Intellectual Disability and Emotional Regulation

According to the American Association on Intellectual and Developmental Disabilities, intellectual disability (ID) is defined as significant limitations in intellectual functioning and adaptive behavior that emerge before the age of 18 [13]. Various authors indicate that emotional development plays a crucial role in different adaptive capacities [14,15,16], among other aspects.

It is important to highlight the value of adapting interventions to people with ID. These adaptations are essential to ensure that interventions are matched to the different cognitive and emotional profiles of this population. By tailoring psychological approaches to their specific needs, professionals can provide more meaningful support, enabling people with ID to function more effectively in different aspects of their lives. This individualized approach enables them to develop coping strategies and improve their self-esteem and fosters the acquisition of essential life skills, ultimately promoting their overall psychological well-being. Through this adapted intervention, we can promote an inclusive society that recognizes the potential and strengths of people with ID, enabling them to thrive and make meaningful contributions to their communities.

Self-care plays a crucial role in the recovery from illness or traumatic experiences. However, it is also essential for individuals who require structured habits and routines for optimal functioning. This is particularly relevant for individuals with ID.

People with intellectual disabilities are characterized by limitations in cognitive development and adaptive behaviors, such as conceptual, social, and practical daily living skills [15]. To enhance the quality of life and well-being of individuals with ID, it is necessary to explore and disseminate intervention strategies that promote adaptive capacities through self-awareness and emotional regulation [16]. Several studies have demonstrated the effectiveness of intervention programs to increase and strengthen emotional skills in children and adolescents [17,18,19,20]. However, interventions specifically tailored to individuals with ID are still limited, and existing efforts show promising but preliminary outcomes [21,22,23]. González [21] conducted research with individuals with ID and found improved self-control, emotion regulation, and increased empathy through an emotional regulation promotion program. Positive emotion is a central component of study within the framework of positive psychology, and the promotion of positive emotion is associated with positive outcomes in terms of health and quality of life [24].

Very few studies have focused on examining the impact of self-care behaviors on individuals with disabilities. Therefore, the objective of this research is to fill this gap by offering an adapted protocol of a positive psychological intervention for individuals with intellectual disabilities and analyzing the efficacy of this intervention in a heterogeneous sample (ID—NoID).

## 2. Materials and Methods

This study consists of two related sections. The first section relates to the adaptation process of the positive psychological intervention called “Emotional Styles” for heterogeneous groups of workers that include individuals with ID.

The second section of the study focuses on the implementation of the adapted intervention. The section presents the results and analysis derived from the intervention implementation.

The first section of this study describes the process of adapting the “Emotional Styles” intervention for its application in heterogeneous groups of workers, including those with ID. The adaptation stages are detailed, taking into consideration the specific needs and characteristics of this group of workers, as well as the incorporation of Easy Read strategies [25] to ensure comprehension and full participation.

This adaptation of the Emotional Styles intervention was carried out in a foundation that is committed to offering care, training, resources, and integration opportunities for individuals with ID in the workforce. Currently, this foundation has over 840 employees. The organization’s motto is social inclusion, and its success is built upon its human capital. Consequently, the organization has a systematic and specific training plan in place for its members.

The initial step prior to conducting this intervention was an assessment of psychosocial factors using the HERO methodology [26]. The HEROCheck questionnaire was administered to all employees to gather baseline information for the intervention program.

Based on the results of the HERO psychosocial assessment, in collaboration with the organization, the target groups most susceptible to receiving this type of intervention were determined.

Furthermore, to facilitate the application for people with special needs, specifically for individuals with functional diversity, the evaluation questionnaire was adapted to the Easy Read methodology by professionals in the field.

Additionally, the Emotional Styles intervention protocol was adapted. This adaptation took into account the content, materials, and timing of the protocol. It was conducted both on site and between sessions to maximize the effectiveness of the intervention. Support personnel for individuals with ID were involved to help with their participation in different intervention activities. Given the characteristics of the group, several actions were taken, listed below:-Adaptation of the materials and resources used in the intervention sessions, either in the language or in the methodology applied used to explain complex concepts such as neuroplasticity. Likewise, the audiovisual content was adapted by reducing the color ranges of the presentations, the number of words, and the location on the screen.-Adaptation of the evaluation questionnaires to the Easy Reading methodology for people with intellectual disabilities. The sociodemographic data sections were adapted. Additional adaptations were made to the satisfaction survey, the evaluation of the transfer of learning to daily life and work, and the questionnaires. The response time to these questionnaires was extended for people with intellectual disabilities. Also, the response scales of all items were adapted to make it easier for people with special needs to understand.-Adaptation of the practical exercises in the different sessions. This intervention was carried out in person and in a state of health alert due to the COVID-19 pandemic. This reality led to the fact that in the different sessions, we had to use masks, which made the exercises associated with mindfulness (breathing, body scan, etc.) very difficult. Alternative practices, not focused on breathing, were sought to achieve the same effects (mindfulness walking, savoring, etc.). The exercises were also adjusted to group use given that groups responded better to this type of exercise compared with individual ones.-The times of the sessions were also adapted to this group with functional diversity. The sessions did not last more than two hours, with a 15 min break at the end of the hour. In this way it was possible to rest and connect better with the proposed exercises.

The second section of this study consisted of the positive intervention. The intervention was designed longitudinally and applied for six weeks, with a weekly two-hour session. The intervention was conducted by professional researchers specialized in the covered topics. Figure 1 shows the structure of the intervention program, the contents covered in each session, and the duration of the workshop.

The intervention consisted of six weekly sessions, each lasting two hours and including a short break. The format was predominantly practical to ensure engagement, especially among participants with intellectual disabilities, who often benefit more from experiential activities than from abstract theoretical content. Sessions 1 and 2 introduced the concept of emotional styles and their neuroscientific foundations, focusing specifically on the dimensions of attention and self-awareness. Sessions 3 and 4 explored resilience and outlook, while sessions 5 and 6 addressed context sensitivity and social intuition. Across all sessions, participants engaged in simplified theoretical discussions, mindfulness-based practices (such as savoring or walking meditation), and group exercises aimed at enhancing emotional understanding. For individuals with intellectual disabilities, materials were adapted to Easy Read standards, presentations were visually adjusted (e.g., simplified layouts and colors), and support staff were present to facilitate participation. COVID-19 safety measures were observed throughout the intervention, and alternative activities were used in place of breath-based mindfulness when required.

The duration of six weekly sessions was selected to ensure a balance between scientific rigor, learning efficacy, and organizational feasibility. From a practical standpoint, the intervention needed to integrate smoothly into the foundation’s workflow without overloading participants, many of whom had limited attentional capacity or required structured routines. The six-session format allowed each emotional dimension to be addressed in depth while maintaining participants’ engagement and minimizing cognitive fatigue. This decision was also grounded in empirical evidence showing that short-format interventions—when well-structured—can generate significant and lasting improvements in psychological well-being and emotional regulation [5,27].

### 2.1. Method

The research design employed in this study is a longitudinal design with a pretest, intervention, post-test, and follow up (6 months after the program is finished). Due to the specific context in which the intervention took place, it was not feasible to randomize the intervention (Figure 2).

Although groups were not matched on sociodemographic variables, preliminary ANOVA analyses indicated no significant differences in gender or age distribution between the ID and non-ID groups. However, differences in literacy levels and support received during the intervention may have influenced outcomes and are acknowledged as potential confounds.

#### 2.1.1. Sample

For this research, an organization from the service sector was selected. This organization employs individuals with diverse functional abilities, predominantly individuals with intellectual disabilities. It has a total of 848 employees, with women comprising 65% of the workforce.

Prior to the intervention, a psychosocial risk assessment was conducted across the organization using the HEROCheck diagnostic tool [26]. Based on these results, the human resources department, in collaboration with the research team, identified the most appropriate work units and departments to receive the intervention, prioritizing those showing lower levels of well-being and higher vulnerability. Within these units, a non-probabilistic convenience sampling strategy was applied: participation in the program was voluntary, and employees were invited to enroll based on their availability and willingness to commit to the full six-week program. Due to the limited size of the eligible population and operational constraints, a priori power estimation was not performed.

The sample size was determined based on the organizational constraints and the voluntary participation of employees in the training program. Given the limited availability of workers and the requirement to ensure proportional representation of individuals with and without intellectual disabilities, a non-probabilistic convenience sampling strategy was used. The total number of participants (N = 45) reflects the maximum feasible cohort that could complete the full six-week intervention during the scheduled period. This approach is consistent with other inclusive intervention studies in real-world organizational settings [28,29].

Regarding the implementation of the positive psychological intervention program, a sample of 45 individuals participated, including 12 individuals with diverse functional abilities (26.66%). Among the participants, 64.4% were women. The ages of the participants in this sample ranged from 19 to 61 years. Regarding age distribution, the majority of individuals fall within the 18–35 age range (38%), followed by 36–45 (35%) and 45–61 (27%) age ranges.

Participants with intellectual disabilities (IDs) were identified by the organization’s human resources department, based on their official certification of disability issued by relevant public authorities. All participants with ID were legally recognized as having significant limitations in intellectual functioning and adaptive behavior before the age of 18, in accordance with international standards such as those proposed by the American Association on Intellectual and Developmental Disabilities (AAIDD) [30]. These individuals were already integrated into supported employment programs within the organization.

In terms of job functions, participants were employed in a wide range of roles within the organization. Employees with intellectual disabilities performed tasks related to gardening; catering services; kitchen support; front-desk reception; laundry services; and, in some cases, coordination of small teams. These roles were adapted to their individual capabilities and supported by job coaches or specialized personnel when needed. Non-ID participants held positions across administrative, technical, and service-related areas. Although all participants worked within the same institutional framework, differences in job complexity, required qualifications, and access to support resources were present and may have influenced engagement with the intervention.

#### 2.1.2. Measures

The intervention incorporated the following variables and measurement tools:

Emotional Styles were assessed using the Emotional Styles Questionnaire (ESQ) [11]. The ESQ is an 18-item self-report questionnaire that captures individual variations in the six dimensions considered crucial for a healthy and happy emotional life. Participants rated their responses on a 7-point Likert scale, indicating their level of agreement (1 = strongly disagree; 7 = strongly agree).

Well-being was measured using the Workplace PERMA Profiler [31]. This comprehensive tool assesses well-being across five dimensions based on the PERMA model: positive and negative emotion, engagement, relationships, meaning, accomplishment, and health. Participants provided ratings on an 11-point Likert scale, indicating the extent to which each statement applied to them (0 = not at all; 10 = completely) (α = 0.86).

Work engagement was assessed using the Ultrashort Utrecht Work Engagement Scale (UWES-3) [32]. The UWES-3 captures the three characteristic dimensions of work engagement: dedication, absorption, and vigor. Participants rated their responses on a 7-point Likert scale, ranging from 0 (strongly disagree) to 6 (strongly agree) (α = 0.84).

Satisfaction level was assessed using a self-constructed questionnaire. This scale was constructed to assess the participants’ satisfaction level with the training received. Participants rated their responses on a 5-point Likert scale, where 1 signifies extremely dissatisfied and 5 signifies extremely satisfied with the intervention. Satisfaction was evaluated only at T2 at the end of the intervention.

It is worth noting that the measurement instruments were adapted by professionals in the field to the Easy-to-Read [25] style for individuals with intellectual disabilities.

Inclusion criteria required participants to be currently employed at the organization, aged 18 or older, and available to attend all six intervention sessions. Participants who failed to complete at least 80% of the sessions or who withdrew informed consent at any point during the process were excluded from the final analysis.

#### 2.1.3. Data Analysis

First, descriptive analyses were conducted with the study variables. Then, one-factor analysis of variance (ANOVA) was applied using the IBM SPSS 26 program [33], to examine if there were significant differences between the group with ID and the group without intellectual disabilities (NoID) before the intervention took place. Second, to test the effects of the intervention program, data were analyzed with repeated-measures ANOVA consisting of one between-subjects factor (ID with or NoID) and one within-subjects factor (time: pre intervention test (T1), post intervention test (T2), and follow-up (FUP)). These analyses were carried out in order to observe differences in the means of each variable depending on the group. The effect represented by the time factor (T1, T2, and FUP) would show whether the Emotional Styles protocol was effective from a general approach, whereas the effect obtained according to the group (ID or NoID) would show whether there were differences between the groups at the level of the general mean. Finally, interaction effects were examined by comparing time factors (T1, T2, and FUP) across each group (ID and NoID). A significance level of 0.05 was established for all the tests.

Despite the small sample size, parametric statistics were applied based on the robustness of repeated-measures ANOVA under moderate sample conditions, as supported in the previous methodological literature [34]. Assumptions of normality were tested using the Shapiro–Wilk test, and no significant deviations were found in the main variables. Nonetheless, the limited statistical power is acknowledged as a limitation in the Discussion Section.

## 3. Results

First, Table 1 shows internal consistencies (Cronbach’s α) between the questionnaires for T1, T2, and FUP scores for the whole intervention group (ID and NoID, N = 45).

Next, Table 2 shows the mean and standard deviations for T1, T2, and FUP scores for all the variables for the whole intervention group (ID and NoID, N = 45). Finally, we tested whether there were significant differences between ID and NoID on the demographic variables before the intervention (pre-time). One-factor ANOVA results indicated no differences between the two groups on the demographic data and gender. With these results, we proceeded to carry out the study, concluding that the two groups were comparable.

The results suggest significant changes in groups across time (T1, T2, and FUP), across groups (ID and NoID), and across the time × group interaction.

Taking into account the time (T1, T2, and FUP), a repeated-measure ANOVA for each of the variables showed statistically significant differences in the following variables:

First, statistically significant differences were found in ESQ outlook measures at the three time points (T1, T2, and FUP) with a large effect size, F(1.51) = 25.57, *p* < 0.001, η2 = 0.516, 1 − β = 0.1, where the scores at T2 (M = 4.95, SD = 1.18) were higher and more statistically significant than at T1 (M = 3.92, SD = 1.09). Additionally, the FUP results (M = 2.66, SD = 1.22) were lower and the difference was statistically significant compared to T1 more (Figure 3).

Second, statistically significant differences were found in ESQ resilience measures at the three time points (T1, T2, and FUP) with a large effect size, F(1.89) = 20.61, *p* < 0.001, η2 = 0.462, 1 − β = 0.1, where the scores at FUP (M = 3.07, SD = 1.25) were lower and the difference was statistically significant compared to T1 (M = 4.4, SD = 0.95). Additionally, the FUP results were lower and more statistically significant than at T2 (M = 4.64, SD = 1.39) (Figure 4).

Third, statistically significant differences were found in PERMA engagement measures at the three time points (T1, T2, and FUP) with a large effect size, F(2) = 9.39, *p* < 0.005, η2 = 0.281, 1 − β = 0.932, where the scores at T2 (M = 7.81, SD = 1) were higher and the difference was statistically significant compared to T1 (M = 7.01, SD = 1.29). Similarly, the FUP results (M = 8.02, SD = 1.37) were higher and more statistically significant than at T1 (Figure 5).

Furthermore, statistically significant differences were found in PERMA relationship measures at the three time points (T1, T2, and FUP) with a large effect size, F(1.34) = 38.83, *p* < 0.001, η2 = 0.347, 1 − β = 0.972, where the scores at T2 (M = 7.96, SD = 1.47) were higher and the difference was statistically significant compared to T1 (M = 6.63, SD = 1.75). Similarly, the FUP results (M = 8.06, SD = 1.28) were higher and more statistically significant than at T1 (Figure 6).

Finally, statistically significant differences were identified in PERMA negative emotion measures at the three time points (T1, T2, and FUP) with a large effect size, F(1.36) = 22.46, *p* < 0.001, η2 = 0.484, 1 − β = 0.999, where the scores at T2 (M = 3.66, SD = 2.09) were lower and the difference was statistically significant compared to T1 (M = 5.67, SD = 2.3). Additionally, the FUP results (M = 4.09, SD = 2.4) were lower and the difference was statistically significant compared to T1 (Figure 7).

Taking into account the groups (ID—NoID), a repeated-measure ANOVA for each of the variables showed statistically significant differences between groups of belonging on ESQ outlook, F(2) = 4.74, *p* < 0.05, η2 = 0.143, 1 − β = 0.66 (Figure 8); ESQ resilience, F(2) = 6.19, *p* < 0.005, η2 = 0.212, 1 − β = 0.87 (Figure 9); ESQ context sensibility, F(2) = 14.2, *p* < 0.001, η2 = 0.382, 1 − β = 0.99 (Figure 10); PERMA happiness, F(2) = 4.1, *p* < 0.05, η2 = 0.151, 1 − β = 0.69 (Figure 11); and PERMA meaning, F(2) = 5.92, *p* ≤ 0.005, η2 = 0.205, 1 − β = 0.85 (Figure 12).

First, the results indicate that the ESQ outlook variable from the NoID group had a significant increase from T1 to T2 when compared with the ID group. Moreover, the NoID group had a significantly higher decrease in FUP than the ID group. Second, the ESQ context sensibility variable had similar results in both groups from T1 to T2, despite the ID group having a significantly positive effect at FUP and the NoID group having a significant decrease. Third, there was a better implementation of the ESQ resilience variable in the NoID group than the ID group. However, both groups experienced a decrease in their results at FUP. Fourth, the PERMA happiness variable tended to increase in both groups. However, the NoID group maintained its tendency to improve at FUP, and the ID group suffered a decline. Finally, the PERMA meaning variable had better results in the NoID group given how its growth was sustained from T1 to T2 and finally at FUP. In the ID group, a decrease in the score is observed, both from T1 to T2 and at FUP.

This figure shows the mean scores on the “outlook” dimension across three time points, pre-intervention (T1), post-intervention (T2), and 6-month follow-up (FUP), comparing participants with intellectual disabilities (ID) and those without (NoID). Higher scores indicate a more optimistic and sustained positive emotional response style. The NoID group showed a strong initial increase in outlook scores after the intervention (T2), followed by a decline at follow-up. In contrast, the ID group showed a more modest but stable improvement across all three time points.

Mean scores on the “resilience” dimension at T1, T2, and FUP were analyzed for ID and NoID participants. Resilience refers to the ability to recover from negative emotions. The NoID group showed a post-intervention peak followed by a significant decline at follow-up, whereas the ID group showed minor improvement followed by stabilization (Figure 9).

Context sensitivity captures the ability to adapt emotional responses to the social environment. The ID group showed a steady and meaningful improvement over time, while the NoID group’s scores peaked post-intervention but declined at follow-up (Figure 10).

This figure displays self-reported happiness levels over time. The NoID group experienced increasing happiness through all three time points. In contrast, the ID group showed improvement at T2 but a decrease by FUP (Figure 11).

Perception of meaningfulness at work increased notably for the NoID group and remained stable at follow-up. The ID group showed lower overall levels and a slight decrease over time (Figure 12).

Mean engagement scores across the three time points were analyzed. Engagement refers to being deeply involved and absorbed in one’s work. Both groups showed improvements post-intervention, with the NoID group showing greater gains. Improvements were sustained at follow-up (Figure 13).

Average scores on the quality of interpersonal relationships were analyzed. Both groups reported significant improvement after the intervention, with gains maintained or slightly increasing at follow-up (Figure 14).

This figure reflects changes in the frequency of negative emotional experiences (reverse-coded). Both groups showed a reduction in negative emotions at T2, with partial rebound at follow-up, more marked in the NoID group.

Attention refers to the capacity to stay focused. The NoID group showed consistent improvements, while the ID group maintained similar levels across all three time points, with minimal variation (Figure 15).

Taking into account the time × group interaction, a repeated-measure ANOVA for each of the variables revealed statistical significance in context sensibility, F(1) = 684, *p* < 0.001, η2 = 0.967, 1 – β = 1 (Figure 10), and attention, F(1) = 684, *p* < 0.025, η2 = 0.634, 1 − β = 1 (Figure 16, Table 3).

Finally, the satisfaction of the 45 intervention participants was assessed. Their satisfaction level reached 100%, with 93% reporting being very satisfied or extremely satisfied. The remarkable 100% satisfaction rate observed among participants in the emotional styles intervention suggests that the program effectively addressed a spectrum of emotional needs. This high level of satisfaction may be attributed to the intervention’s adaptability, resonating with the varied emotional experiences of the participants.

## 4. Discussion

This study aimed to adapt and implement a positive psychological intervention based on the Emotional Styles model within an inclusive workplace context, assessing its impact on psychological well-being and stress in employees with and without intellectual disabilities (IDs). The findings offer robust empirical evidence supporting the effectiveness of such interventions in fostering emotional resources and mitigating distress, in both the short and the medium term.

First, analyses revealed significant improvements in key variables related to well-being, particularly outlook, resilience, engagement, relationships, and the reduction in negative emotions. These dimensions are strongly linked to emotional regulation, understood as a process that facilitates adaptive responses to environmental demands [1,2,3,4,5,6,7,8,9,10,11,12,13,14,15,16,35,36,37]. The observed effects empirically validate the potential of the neuropsychological approach to emotional styles as a foundation for applied interventions beyond laboratory contexts and among diverse populations.

Notably, sustained improvements were found in engagement and relationships, dimensions that directly influence positive work experiences and team cohesion. These findings align with previous research on the effectiveness of positive interventions in organizational settings [5,6,7,8,9] and reinforce the role of emotional training in promoting healthy, resilient, and inclusive work environments [28].

In addition, the significant reduction in negative emotions suggests a protective effect against stress and psychological discomfort. Given that this variable is typically resistant to change in brief interventions, its decrease at both post-intervention and follow-up stages is particularly promising. It may indicate that even adapted training in emotional awareness and mindfulness-based techniques was sufficient to produce lasting improvements in emotional regulation.

Group-based analysis provided relevant nuances. The non-ID group showed greater sustained benefits in variables such as resilience, outlook, happiness, and meaning, whereas the ID group showed more moderate gains or even declines at follow-up. This divergence may be explained by several factors, including differences in the transferability of skills to daily life, cognitive limitations in generalizing emotional learning, or limited access to ongoing support outside the intervention setting. However, it is worth highlighting that context sensitivity showed maintained or even enhanced improvement within the ID group, suggesting that certain social-interactional skills may be especially responsive to adapted interventions.

These group differences also prompt reflection on the adaptation process itself. Despite considerable efforts to incorporate Easy Read strategies and adjust content to the cognitive profiles of participants with ID, future interventions may require extended formats, more frequent reinforcement, and structured post-intervention follow-up to consolidate emotional learning effectively particularly considering the persistent organizational barriers identified in the prior literature [38].

Regarding satisfaction, 100% of participants expressed overall satisfaction with the experience, with 93% reporting being very or extremely satisfied. This result not only confirms the relevance of the content and delivery but also underscores the sensitivity and inclusiveness with which the intervention was implemented. Such satisfaction is particularly noteworthy given the contextual challenges posed by the COVID-19 pandemic, which significantly affected logistical and interpersonal aspects of the program.

Taken together, these findings support the conclusion that adapted, context-sensitive, and sustained positive psychological interventions can serve as effective tools for enhancing emotional well-being in diverse work settings. While preliminary, the findings suggest that adapted positive psychological interventions may be beneficial for supporting the emotional well-being of employees with intellectual disabilities in inclusive work settings. These results should be interpreted cautiously, and future research with more robust designs is needed to confirm the potential for long-term empowerment and inclusion.

## 5. Limitations and Future Research

This project is situated within a complex social, economic, and health-related period. Alongside the socio-economic challenges inherent in recovering from the global crisis, there are also those brought about by the COVID-19 pandemic. The working environment has been greatly impacted by new forms of work and changes in the production processes. All of this has negatively affected individuals’ overall health, both in their professional and personal lives. In relation to the project’s implementation, several obstacles have been identified, most of which are related to the pandemic-induced working conditions. Fortunately, all of these obstacles have been successfully overcome.

The intervention design did not include randomization or a control group, which limits the causal strength of the findings. The results should be interpreted as exploratory and context-dependent. Future studies should incorporate randomized controlled trials and larger samples to assess effectiveness more rigorously.

First, selecting the organization in which to carry out the intervention and psychological project was a challenge. Identifying organizations with a presence of employees with functional diversity was difficult as finding a minimum percentage of workers with these characteristics is hard to come by.

Second, adapting the protocol and materials to the groups’ characteristics posed a challenge due to the lack of precedents in this regard. This required adapting the workshop’s execution almost in real time. The materials and tools that were used demanded significant effort, which was carried out under considerable time pressure.

Third, due to the pandemic’s working conditions, the intervention was initially planned to be conducted online. However, considering the attendees’ characteristics, it was reformulated to be carried out in person. This required the organization’s effort in finding a sufficiently large space to accommodate the group of attendees while adhering to COVID-19 safety measures.

Fourth, the small sample size limits statistical power, and although parametric tests were applied with appropriate diagnostics, future studies with larger samples are recommended to confirm these results.

Additionally, differences in support structures and literacy levels between groups may have influenced how participants engaged with the intervention content. These factors were not systematically controlled and should be taken into account in future research using more refined group-matching or stratified designs.

Last, unforeseen difficulties arose during the statistical analysis process. The obtained results, while intriguing, refer to a relatively small sample size, and as such, they must be interpreted within this context.

A promising area of research would be to explore the transferability of the intervention to other populations with functional diversity. Investigating how the adaptation could benefit different groups of workers with various abilities and challenges could provide valuable insights for expanding the intervention at a community level.

Furthermore, underscoring the need for a randomized controlled trial is paramount. Implementing this methodological approach in future research would ensure a more robust evaluation of the intervention’s impact, establishing a solid evidence base for its effectiveness in comparison with alternative interventions.

## 6. Conclusions

This study provides preliminary evidence for the effectiveness and feasibility of adapting positive psychological interventions for inclusive workplace environments. Despite limitations, the findings suggest that brief, neuropsychologically grounded programs can foster emotional well-being among diverse employee populations. Future research should prioritize methodological rigor and explore long-term transferability to varied organizational contexts.

## Figures and Tables

**Figure 1 healthcare-13-02096-f001:**
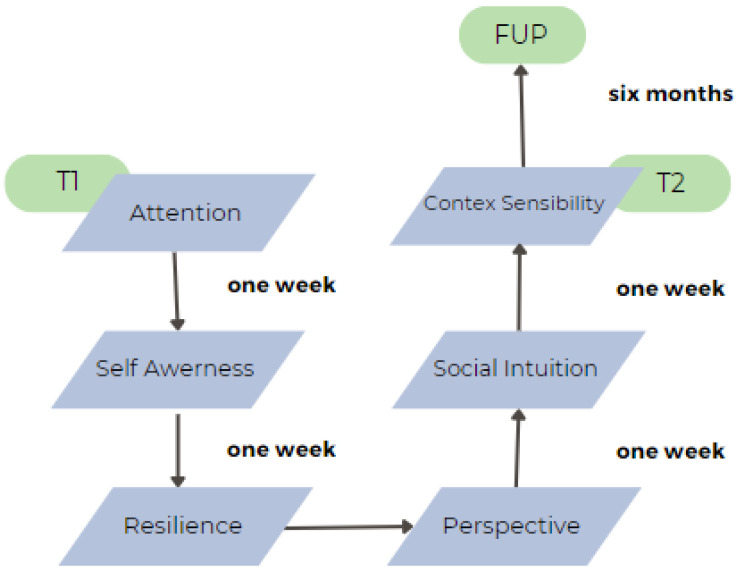
Emotional Styles program (N = 45).

**Figure 2 healthcare-13-02096-f002:**
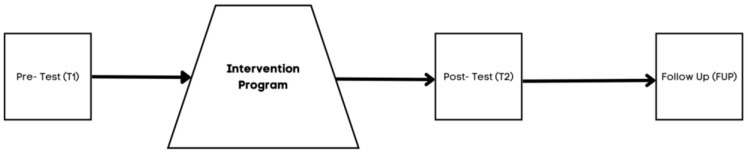
Design of the study.

**Figure 3 healthcare-13-02096-f003:**
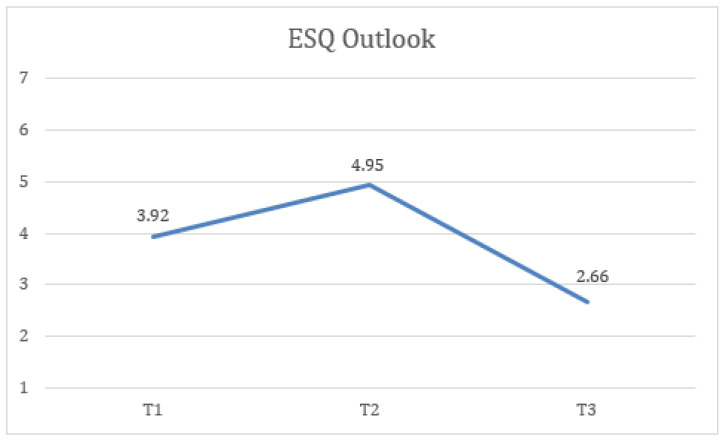
Change in outlook (ESQ) scores over time for the total sample.

**Figure 4 healthcare-13-02096-f004:**
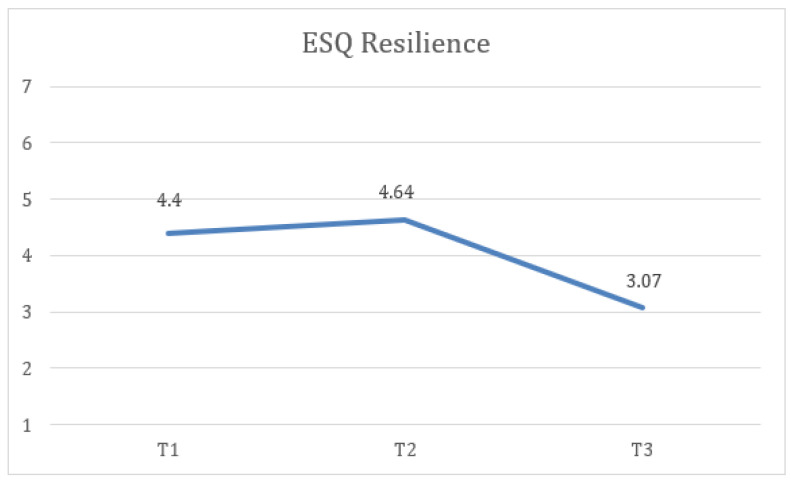
Change in resilience (ESQ) scores over time for the total sample.

**Figure 5 healthcare-13-02096-f005:**
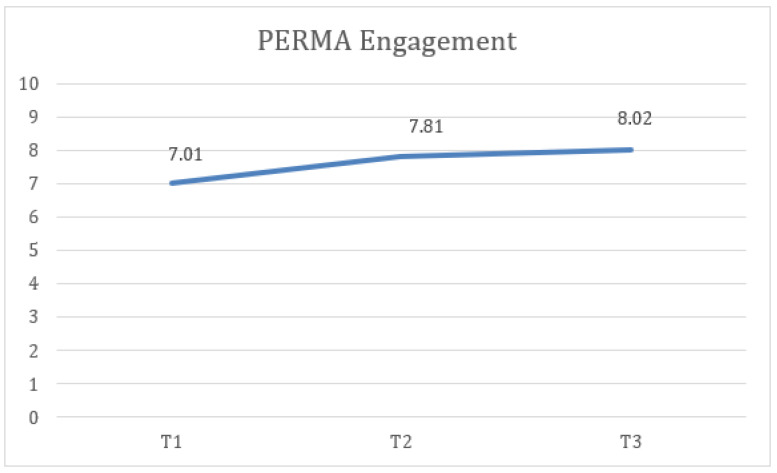
Change in engagement (PERMA) scores over time for the total sample.

**Figure 6 healthcare-13-02096-f006:**
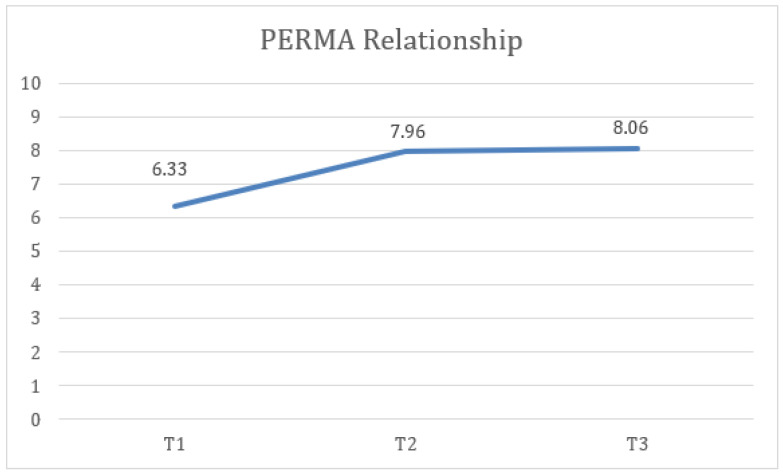
Change in relationship (PERMA) scores over time for the total sample.

**Figure 7 healthcare-13-02096-f007:**
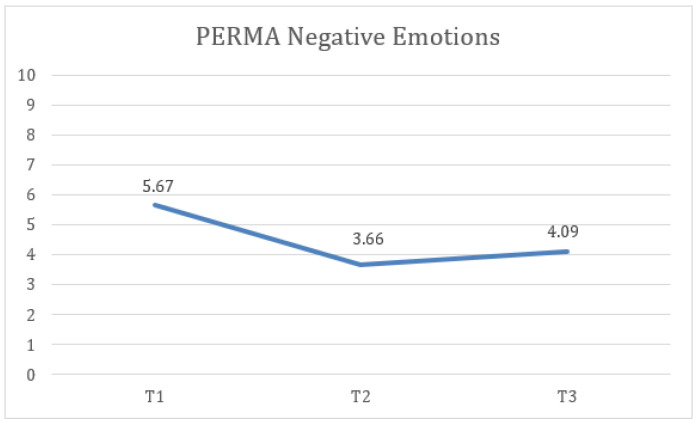
Change in negative emotion (PERMA) scores over time for the total sample.

**Figure 8 healthcare-13-02096-f008:**
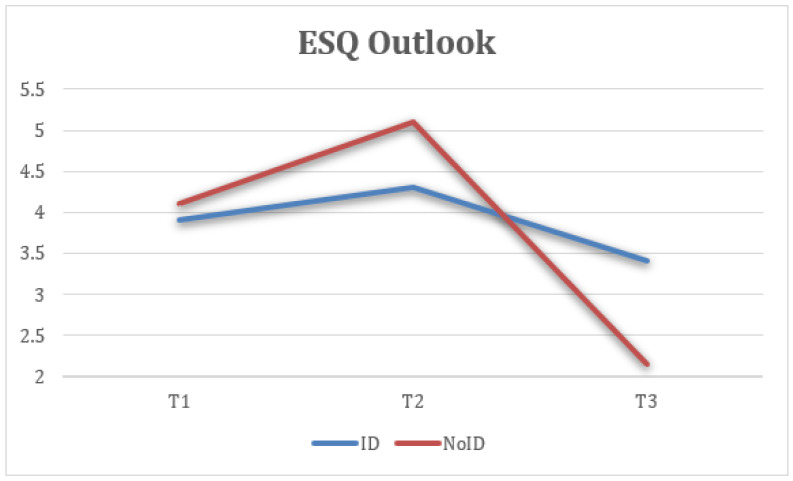
Interaction between time and group on outlook (ESQ).

**Figure 9 healthcare-13-02096-f009:**
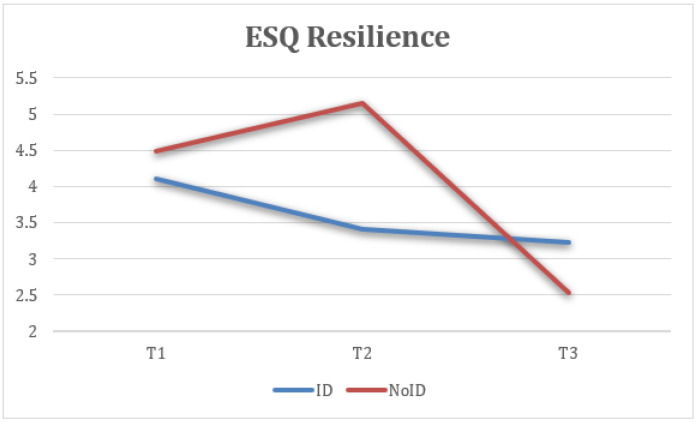
Interaction between time and group on resilience (ESQ).

**Figure 10 healthcare-13-02096-f010:**
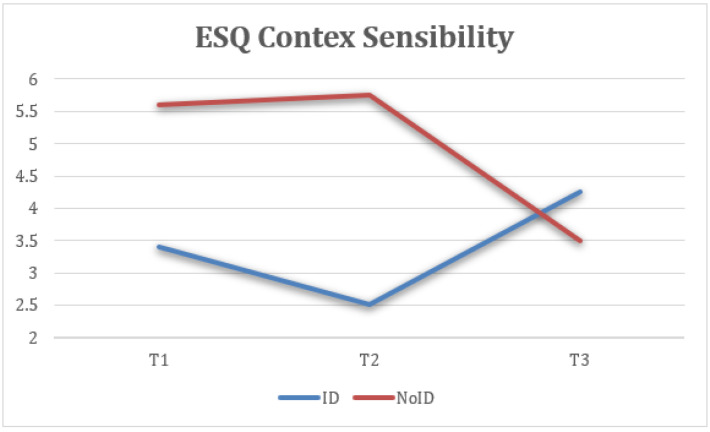
Interaction between time and group on context sensitivity (ESQ).

**Figure 11 healthcare-13-02096-f011:**
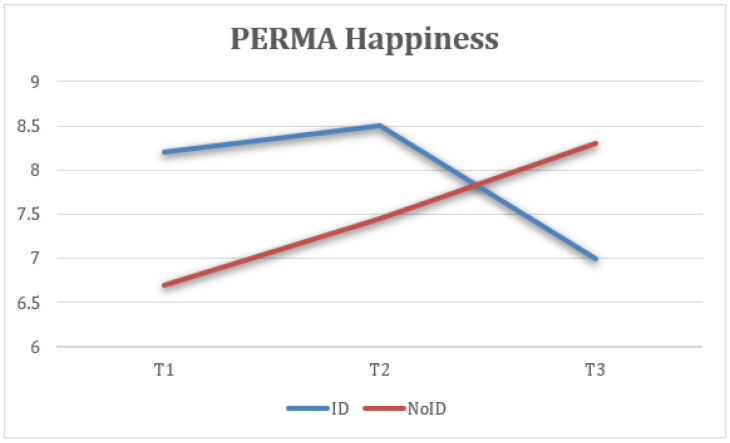
Interaction between time and group on happiness (PERMA).

**Figure 12 healthcare-13-02096-f012:**
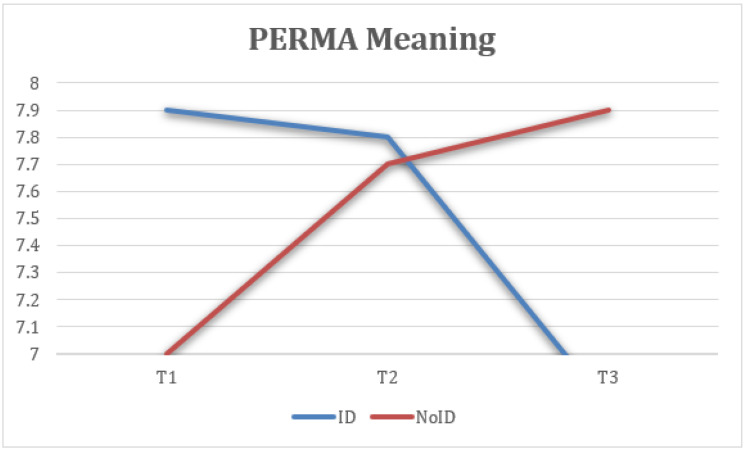
Interaction between time and group on meaning (PERMA).

**Figure 13 healthcare-13-02096-f013:**
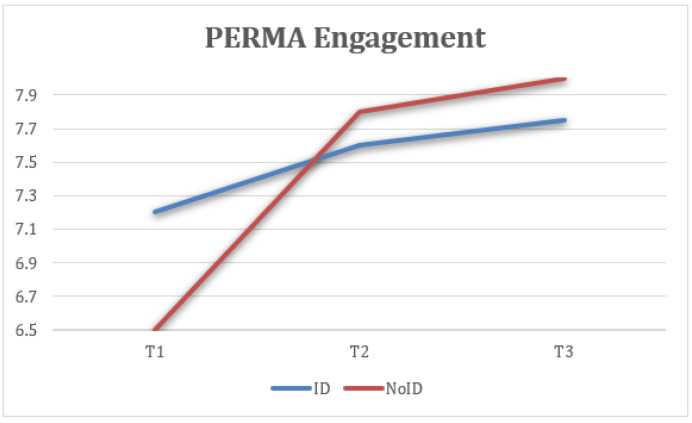
Interaction between time and group on engagement (PERMA).

**Figure 14 healthcare-13-02096-f014:**
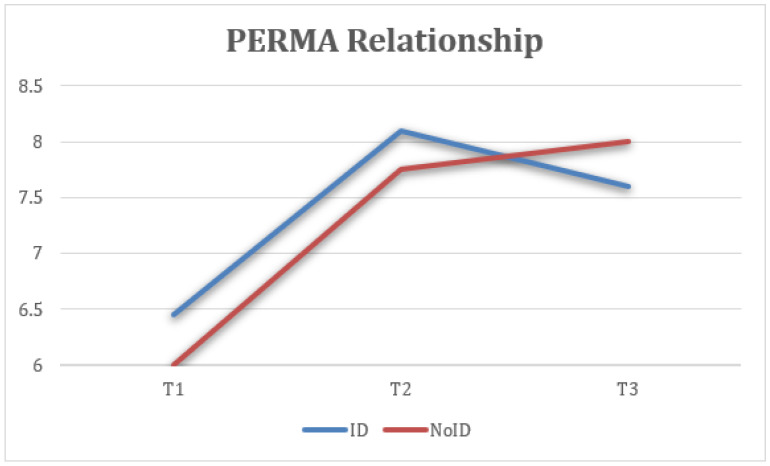
Interaction between time and group on relationship (PERMA).

**Figure 15 healthcare-13-02096-f015:**
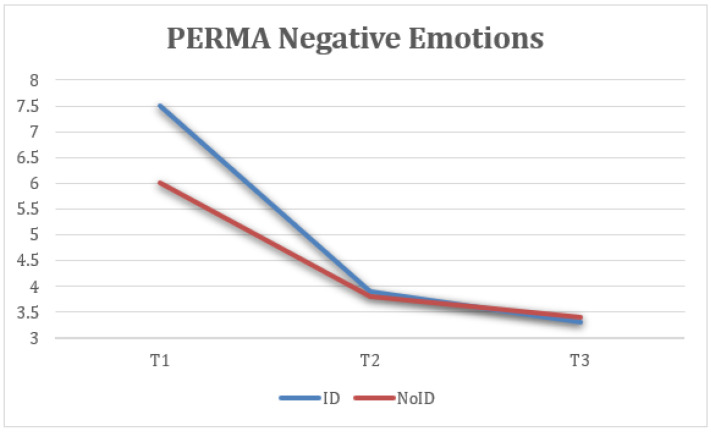
Interaction between time and group on negative emotions (PERMA).

**Figure 16 healthcare-13-02096-f016:**
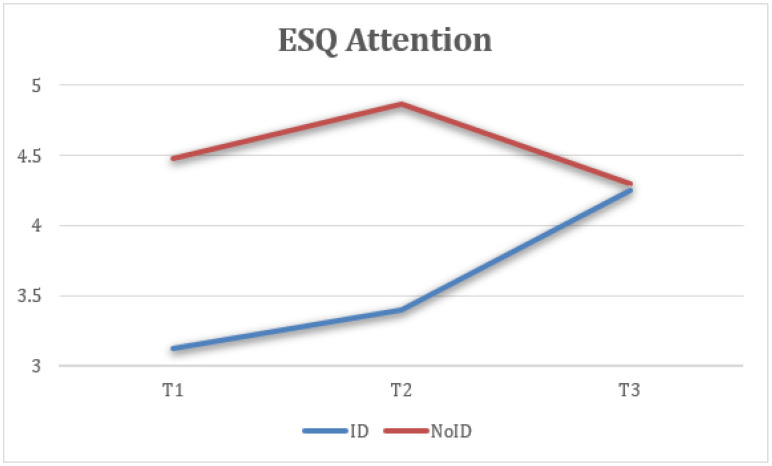
Interaction between time and group on attention (ESQ).

**Table 1 healthcare-13-02096-t001:** Internal consistencies for the questionnaires for the whole intervention group (N = 45).

Measures	α	α	α
	T1	T2	T3
1. UWES-3 Engagement	0.82	0.71	0.69
2. ESQ Emotional Style	0.66	0.70	0.83
3. PERMA	0.79	0.80	0.87

Notes: ESQ—Emotional Styles Questionnaire; PERMA—Positive Emotion, Engagement, Relationships, Meaning, and Accomplishment; UWES-3—Utrecht Work Engagement Scale (three-item version). α = Cronbach’s alpha coefficient indicating internal consistency reliability. Higher α values (>0.70) indicate acceptable reliability.

**Table 2 healthcare-13-02096-t002:** Pre-, post-, and FUP intervention means and standard deviations for all the variables for the whole intervention group (N = 45).

Variables	M T1	SD	M T2	SD	M FUP	SD
1. UWES 3 Engagement	4.68	1.04	5.15	0.84	5.27	0.82
2. EE Outlook	3.92	1.09	4.95	1.18	2.66	1.22
3. EE Resilience	4.40	0.95	4.64	1.39	3.07	1.25
4. EE Self Awareness	4.34	1.14	4.51	0.77	4.35	0.74
5. EE Sensibility to Context	5.07	1.46	4.90	1.75	3.96	1.06
6. EE Social Intuition	4.23	1.26	5.07	0.89	3.47	0.87
7. EE Attention	4.20	1.26	4.45	1.31	4.42	0.68
8. PERMA Engagement	7.01	1.29	7.81	1.00	8.02	1.37
9. PERMA Relationships	6.63	1.75	7.96	1.47	8.06	1.28
10. PERMA Meaning	7.29	1.20	7.79	1.23	7.65	1.40
11. PERMA Accomplishment	7.11	1.25	7.50	1.32	7.55	1.46
12. PERMA Health	6.48	1.79	6.49	1.97	7.15	1.98
13. PERMA Happiness	7.17	1.66	7.65	1.75	7.81	1.76
14. PERMA Positivity	7.24	1.41	7.52	1.44	7.61	1.51
15. PERMA Negative Emotions	5.67	2.3	3.66	2.09	4.09	2.4
16. PERMA Loneliness	3.35	2.41	3.37	2.88	3.40	3.18

Notes: T1—pre-intervention; T2—post-intervention; FUP—follow-up (6 months after the intervention). Scores reflect mean values for each dimension measured. Higher scores indicate greater presence of the corresponding emotional or well-being trait.

**Table 3 healthcare-13-02096-t003:** Repeated measures ANOVA’s for the effects of time, group, and time X group X variable interaction.

	Time				Group				Time × Group			
	df Effect	F	*p*	η2	df Effect	F	*p*	η2	df Effect	F	*p*	η2
UWES 3 Engagement	2	523	0.596	0.022	2	4.43	0.017	0.162	1	0.022	0.884	0.052
EE Outlook	2	12.63	<0.001	0.355	2	3.83	0.029	143	1	0.018	0.895	0.052
EE Resilience	2	10.59	<0.001	0.304	2	6.8	0.004	0.212	1	3.153	0.089	0.398
EE Self-Awerness	2	0.287	0.752	0.012	2	2.13	0.130	0.085	1	0.638	0.433	0.119
EE Contex Sensibility	2	1.40	0.255	0.058	2	14.20	<0.001	0.382	1	20.826	<0.001	0.992
EE Social Intuition	2	9.96	<0.001	0.302	2	1.23	0.301	0.051	1	1.132	0.298	0.175
EE Attention	2	0.845	0.436	0.035	2	2.75	0.074	0.107	1	5.781	0.025	0.634
PERMA Engagement	1.6	4.53	0.024	0.165	1.61	0.860	0.401	0.036	1	0.000	0.994	0.050
PERMA Relationships	1.43	8.12	0.003	0.261	1.43	0.284	0.681	0.086	1	0.033	0.857	0.054
PERMA Meaning	2	5.92	0.005	0.045	2	5.92	0.005	0.205	1	0.094	0.762	0.060
PERMA Accomplishment	2	0.402	0.672	0.017	2	1.02	0.238	0.061	1	0.198	0.661	0.071
PERMA Health	2	2.03	0.14	0.081	2	2.43	0.098	0.096	1	0.480	0.495	0.102
PERMA Happiness	2	0.439	0.647	0.019	2	4.10	0.023	151	1	0.779	0.386	0.033
PERMA Positivity	1.72	0.170	0.844	0.007	1.72	0.627	0.539	0.027	1	0.042	0.839	0.054
PERMA Negative Emotions	1.51	21.53	<0.001	0.484	1.51	1.218	0.295	0.050	1	0.382	0.543	0.091
PERMA Loneliness	1.65	2.45	0.097	0.096	1.65	1.533	0.230	0.062	1	0.026	0.873	0.053

Notes: F—ANOVA test statistic; η^2^ = partial eta squared (effect size); *p* < 0.05 indicates statistical significance. Time—three repeated measures (T1, T2, and FUP); group—participants with intellectual disability (ID) and without (NoID). Interaction effects reflect the combined influence of time × group on the dependent variables.

## Data Availability

The data used and analyzed during the current study are available from the corresponding author upon reasonable request.

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
