# Peer review of "Adapting a Positive Psychological Intervention for Employees with and Without Intellectual Disabilities"

_healthcare, 2025, doi:10.3390/healthcare13172096_

Round 1

Reviewer 1 Report

Comments and Suggestions for Authors

Dear authors, 

please elaborate on explaining how the sample size was obtained.

Also, how were the individuals with intellectual disability identified?

Why was the duration of intervention chosen as such?

Please find relevant citations of articles in English.

Please apply the same citation manner throughout the text (e.g. line 125, 387, 394).

Author Response

Dear Reviewer

We sincerely thank you for your insightful feedback. All our responses and the corresponding revisions are presented in detail in the attached response document.

Reviewer 2 Report

Comments and Suggestions for Authors

This manuscript presents a timely and well-executed contribution to the field of workplace well-being and inclusive psychological interventions. One of its key strengths is the innovative adaptation of a positive psychological intervention specifically designed to include individuals with intellectual disabilities, a population that is often underrepresented in organizational well-being research. The topic is both relevant and socially significant, addressing issues of emotional health and inclusion within diverse work environments. The study is methodologically sound, with clear documentation of the adaptation process, the use of validated instruments, and appropriate statistical analyses. Additionally, the high satisfaction levels reported by participants support the acceptability and feasibility of the intervention. Finally, the manuscript offers a valuable discussion of the differing effects across participant groups, enriching the interpretation of the findings and highlighting areas for future research and practical improvement.

While the manuscript possesses notable strengths, some areas need clarification or improvement to boost its overall clarity, methodological transparency, and scientific impact:

  1. Title clarity and accuracy

The current title has grammatical issues and fails to fully convey the scope of the study. The phrase "adaptation to intellectual disability workers" is unclear and should be revised to "workers with intellectual disabilities." Furthermore, the title does not acknowledge the participation of both individuals with and without intellectual disabilities and the workplace context. A more precise alternative could be "Adapting a Positive Psychological Intervention for Employees With and Without Intellectual Disabilities."

2. Figures

While the figures and tables presented are relevant and support the results, many lack detailed captions and clarity. Figures that illustrate interaction effects, such as time × group, would benefit from more informative labels, titles, and explanations in their captions. Additionally, tables summarizing internal consistency or statistical analyses should include notes that explain acronyms, test statistics, and effect sizes. Improving the visual formatting of the figures, such as axis labels, color coding, and font sizes, would greatly enhance their readability and interpretative value.

3. Methodological section:

Lack of Detail on Disabilities: While the study broadly refers to “individuals with intellectual disabilities (ID),” it lacks a clear operational definition or description of the diagnostic criteria, levels of severity, and the methods used to determine ID status. It is unclear whether this status was self-reported, formally diagnosed, or assessed by professionals.

Group Comparison Logic: The intervention compares individuals with and without ID, but it does not specify whether participants were matched on relevant sociodemographic variables or if randomization was attempted (which, as mentioned, was not possible). Additionally, there is limited information on how potential confounding variables (e.g., job roles, support received during the intervention, literacy levels) were controlled or accounted for.

Sample Composition: The sample size is mentioned (N=45, with 12 participants having ID), but there is little context regarding representativeness. For example, what job functions did the participants hold? Were there any differences between the ID and non-ID groups beyond their disability status?

Adaptation Detail vs. Methodological Clarity: Although the adaptations to Easy Read formats are well described (which is a strength of the paper), there is a lack of clarity regarding core methodological decisions, such as sampling strategy, power estimation, and the rationale for inclusion/exclusion criteria.

Comments on the Quality of English Language

The manuscript requires extensive revision and a polished touch to its English language. A comprehensive review is essential to untangle the long and intricate sentences, eliminate redundancies, and refine the technical language that currently disrupts the flow and clarity of the writing.

Author Response

(The authors gave the same response as above.)

Reviewer 3 Report

Comments and Suggestions for Authors

The article presents an experimental research approach within the framework of an intervention, providing relevant contributions applicable to professionals with intellectual disabilities and emotional regulation who need improvement. The research design is, overall, adequate. There are, however, some elements that require more information. Because the research focuses on participants from a population with intellectual disabilities, it is considered that the procedures for approval of the research by an independent ethics committee should be made more explicit, as well as the procedures for obtaining the aforementioned informed consent at the end of the article, given the specificities of the participants and their possible limitation in the capacity to consent because their self-determination may in some cases be conditioned. The presentation of further information on these procedures and elements should condition the decision on approval for publication.

It is considered that the descriptive information on the procedures for implementing the intervention needs to be optimized. A detailed description of the work done should be provided as part of the method.

In section 2.2.3. In data analysis, researchers must provide evidence that data from a small sample can be analyzed using parametric statistics. This element conditions the value of the analyses presented later.

The discussion should be reworded in some sentences to focus on the concrete results obtained, within the context of the experimental limitations. Do not make general statements that are not supported by the results (e.g., “the study provides empirical support for the active inclusion of individuals with intellectual disabilities in organizational well-being programs, moving beyond assistentialist approaches and positioning them as empowered agents of their own psychosocial development”).
The lack of randomization in the constitution of intervention groups and the absence of control groups for comparison are element that significantly detracts from the value and strength of the conclusions of this study. However, the relevance of its approach and the complex circumstances surrounding its development give it value.

Author Response

(The authors gave the same response as above.)
